# The Association of the Publication of a Proposed Public Charge Rule with Preterm Births among Uninsured Foreign-Born Latinx Birthing People in the United States

**DOI:** 10.3390/healthcare11142054

**Published:** 2023-07-18

**Authors:** Sung W. Choi

**Affiliations:** School of Public Affairs, Pennsylvania State University—Harrisburg, 777 West Harrisburg Pike, Middletown, PA 17057, USA; sxc835@psu.edu; Tel.: +1-717-948-6208; Fax: +1-717-948-6320

**Keywords:** public charge rule, preterm birth, maternal and infant health, immigrant health, racial and ethnic health disparities

## Abstract

Following the inauguration, the Trump administration authorized a series of anti-immigrant policies, including modifications to the public charge regulation. This study analyzed the effect of the publication of a proposed public charge rule in 2018 on the risk of preterm birth between uninsured and privately insured Latinx birthing people in the United States by using natality files from the National Center for Health Statistics. In total, 1,375,580 Latinx birthing people reported private insurance as their primary source of delivery from 2014 to 2019, while 317,056 Latinx birthing people reported self-pay as their primary source of delivery during the same period. After the publication of the proposed public charge rule in 2018, the odds of preterm birth among uninsured foreign-born Latinx birthing people increased by 6.2% compared with privately insured foreign-born Latinx birthing people (OR: 1.062; 95% CI: 1.016, 1.110). On the other hand, the odds of preterm births among uninsured US-born Latinx birthing people did not significantly increase after the publication of the proposed rule compared with privately insured US-born Latinx birthing people. These findings suggest the publication of the public charge rule proposed in 2018 may be associated with adverse birth outcomes among uninsured foreign-born Latinx birthing people in the United States.

## 1. Introduction

There was a notable rise in the number of preterm births among Latinx birthing people following the 2016 US presidential election [1]. It was proposed that elevated levels of psychosocial stress and anxiety, which were caused by the anti-immigrant rhetoric during the 2016 US presidential election, could be attributed to the escalating number of premature births among them [2,3]. Furthermore, a greater number of Latinx birthing people either delayed or altogether forewent prenatal care after the 2016 US presidential election. This inadequate prenatal care was also posited as another factor contributing to the heightened risk of premature birth among the Latinx population in the United States [4].

Following the inauguration, the Trump administration authorized a series of executive orders with an anti-immigrant stance, which included measures related to border security, interior enforcement, and curtailing the admission of refugees to the United States (E.O. 13767, 13768, 13769) [5,6,7]. Moreover, the Department of Homeland Security (DHS) implemented extensive modifications to the public charge regulation, which has the potential to considerably limit immigrants’ ability to obtain healthcare services. The public charge rule, which has been an integral part of US immigration legislation since the 1880s, has empowered immigration authorities to deny applications for admission to the United States or lawful permanent residency based on an individual’s past usage of public benefits. The Trump administration broadened the scope of the rule to incorporate cash and noncash benefits, including the Supplemental Nutrition Assistance Program (SNAP), nonemergency Medicaid, Section 8 housing benefits/public housing, and drug subsidies under Medicare Part D [8,9,10]. The changed rule will make public charge determinations by considering not only previous use of public benefits but also the applicant’s potential use of public benefits, which will be evaluated based on the applicant’s personal attributes, such as age, income, education, and English proficiency [9,10]. The changes to the rule were anticipated to make immigrant families avoid public benefits due to immigration-related concerns [9,10].

In January 2017, a draft executive order about the proposed revisions to the public charge rule was divulged. Then, in September 2018, the DHS declared its intention to modify the public charge rule, and the definitive version of the rule was issued in August 2019. Although the public charge rule was temporarily enforced from February 2020 to March 2021, the anticipated implementation of the policy change might have caused noncitizen pregnant people to discontinue their Medicaid coverage, possibly due to the “chilling effect” of the expected modification [9,10].

Previous studies on the changes to the public charge policy analyzed the effects of the changes either on the use of public benefit programs or on adverse mental health outcomes among immigrant families based on survey data [9,10,11,12]. The changes to the public charge rule may engender elevated levels of psychosocial stress and anxiety, which could adversely impact pregnant people in the Latinx community. This study contributes to the body of literature by investigating the effect of the changes to the public charge rule on the likelihood of premature birth based on the entire population of newborns among the Latinx population. The objective of this study is to evaluate effect of the changes to the public charge rule on susceptibility to premature birth among the Latinx population in the United States.

## 2. Methods

### 2.1. Data and Sample

This study primarily utilized the natality file provided by the National Center for Health Statistics (NCHS), which contains various measures related to demographics, socioeconomic status, health utilization, and health outcomes of infants born in the United States. The study sample consisted of all live singletons born in hospitals across 47 states and the District of Columbia between 2014 and 2019 from Latinx birthing people aged 15–44 years. The 2003 US birth certificate revisions were implemented in all 50 states and the District of Columbia starting from 2016. States that had adopted the revisions during the study period, including Connecticut, New Jersey, and Rhode Island, were excluded from the sample.

The study’s treatment group comprises uninsured Latinx birthing people who financed their delivery through “self-pay”, while the comparison group consists of privately insured Latinx birthing people. Uninsured Latinx birthing people were more likely to plan to use Medicaid as the primary sources of delivery than privately insured Latinx birthing people. Those who disenrolled from Medicaid after the changes to the public charge rule were more likely to remain uninsured. Therefore, the treatment group was selected as uninsured Latinx birthing people, because they were more susceptible to the changes to the public charge rule in contrast to their privately insured counterparts.

### 2.2. Outcome Measure, Exposures, and Covariates

The primary outcome measure is the likelihood of preterm birth among Latinx birthing people, which refers to the birth of an infant before the 37th week of the gestational period. The main exposures are the changes to the public charge rule, which include the leaked draft executive order regarding the changes to public charge policy in January 2017 and the publication of a suggested public charge policy in October 2018. This study covers three periods: (1) 1 January 2014 to 31 December 2016; (2) 1 January 2017 to 31 October 2018; and (3) 1 November 2018 to 31 December 2019. This study also controls for other risk factors for preterm birth, such as the birthing people’s age, education, marital status, race/ethnicity, and infant’s birth order. This analysis also controlled for county-level characteristics such as median household income and unemployment rate, as well as state and year fixed effects. Appendix A includes the definitions of the variables, including outcome measures, exposures, and risk factors of preterm birth.

### 2.3. Analysis

This study examined the effect of changes to the public charge rule on preterm birth outcomes among uninsured Latinx birthing people compared with privately insured Latinx birthing people. The analysis employed a difference-in-differences approach that compared the differences in preterm birth before and after the changes to the public charge rule among uninsured Latinx birthing people in relation to privately insured people. Multivariate logistic regression models were utilized in the analysis. The unit of analysis is the birthing person–infant dyad. The results of difference-in-differences analyses for foreign-born Latinx birthing people were compared with the results for US-born Latinx birthing people.

In addition, event studies on preterm birth outcomes were conducted among uninsured Latinx birthing people compared with privately insured people. An event study was selected as the research design of this study because it is more flexible to model time-varying and multiple treatment effects on the outcome measure [13]. This study estimated preterm birth outcome using the following form:(1)$${Outcome}_{ist}=\alpha +\sum_{j=1}^{11}\beta_{j}\cdot \lambda \cdot {Lag\;j}_{t}+\sum_{k=1}^{11}\gamma_{k}\cdot {\lambda \cdot Lead\;k}_{t}+X_{ist}+\mu_{s}+\theta_{t}+\varepsilon_{ist}$$

The following are the definitions of binary lags and leads of the event:$${Lag\;J}_{t}=1\left[t\leq Event-11\right]\;{Lag\;j}_{t}=1\left[t=Event-j\right]\;if\;j\in \;\left\{1,2,3,\;\ldots ,10\right\}\;{Lead\;k}_{t}=1\left[t=Event+k\right]\;if\;k\in \;\left\{1,2,3,\;\ldots ,10\right\}\;{Lead\;k}_{t}=1\left[t\geq Event+11\right]$$

${Lag\;j}_{t}$ are binary lag variables that quarter *t* was *j* periods before the leak of the draft executive order about the changes to the public charge rule, which occurred in January 2017. ${Lead\;k}_{t}$ are binary lead variables that quarter *t* was *k* periods after the leak of the draft executive order in January 2017. The event study includes 11 lag and 12 lead variables during the study period. The lags start from the second quarter of year 2014 (*t* = −11) and end at the last quarter of year 2016 (*t* = −1). The leads start from the first quarter of year 2017 (*t* = 0) and end at the last quarter of year 2019 (*t* = 11). The publication of the proposed public charge rule in October 2018 was coded as *t* = 7. The baseline omitted case is the first quarter of the year 2015, which is *t* = −12.

In the event study, *i*, *s*, and *t* denote individual, county/state, and time identifiers, respectively. *Outcome_ist_* is an indicator variable for preterm birth, λ is an identifier for uninsured Latinx birthing people, and $\beta_{j}$ and $\gamma_{k}$ are parameters that measure the impact of the changes to the public charge rule on uninsured Latinx birthing people compared with privately insured Latinx birthing parent. *X_ist_* is a vector of the birthing person–infant dyad characteristics, including mother’s age, education, marital status, and infant’s birth order and county characteristics, including median household income and unemployment rate. *μ_s_* and *θ_t_* are vectors of state and year fixed effects, and *ε_ist_* is an error term. Stata SE version 16 was used to conduct all statistical analyses, and the Institutional Review Board (IRB) of the author’s university exempted this study from IRB oversight, because this study analyzed publicly available deidentified data (STUDY ID: 00021733).

## 3. Results

### 3.1. Descriptive Statistics

In total, 1,692,636 Latinx birthing people were identified from 1 January 2014 to 31 December 2019. A total of 1,375,580 Latinx birthing people reported private insurance as their primary source of delivery, while 317,056 Latinx birthing people reported self-pay as their primary source of delivery.

Table 1 compares and contrasts the characteristics of infants, birthing people, and preterm birth between the privately insured and uninsured among Latinx birthing people. In addition, the differences between the two groups were further stratified by nativity and the period related to the changes to the public charge rule. Privately insured Latinx birthing people showed significantly different characteristics from uninsured Latinx birthing people. Uninsured Latinx birthing people were more likely to have preterm births, be younger, have lower levels of education, be unmarried, have deliveries of a second-born or more, and live in a county with lower median household income and unemployment rate than privately insured Latinx birthing people.

Foreign-born privately insured Latinx birthing people were more likely to have preterm births compared with US-born privately insured birthing people, while foreign-born uninsured Latinx birthing people were less likely to have preterm births compared with US-born uninsured birthing people. The difference between privately insured and uninsured Latinx birthing people were decreased or did not change over time among most characteristics, including age, education, and county income/unemployment rate. However, the difference in the risk of preterm birth between privately insured and uninsured Latinx birthing people was increased during the period post the leaked draft executive order. After the publication of a proposed public charge rule, the difference in the risk of preterm births between privately insured and uninsured Latinx birthing people became even larger for both US-born and foreign-born Latinx birthing people.

### 3.2. Difference-in-Differences Results

Table 2 reports the association of the publication of the proposed public charge rule in 2018 on the risk of preterm births among the Latinx population in the United States using the difference-in-differences approach. The results for US-born Latinx birthing people are reported in the left column, and the results for foreign-born people are reported in the right column.

The results for US-born Latinx birthing people were quite different from the results for foreign-born Latinx birthing people. Being uninsured was significantly associated with a higher rate of preterm births among US-born Latinx birthing people but not among foreign-born Latinx birthing people. The odds of preterm births significantly increased after the leaked draft executive order among US-born Latinx birthing people but not among foreign-born Latinx birthing people.

We found that the leaked draft and publication of a proposed public charge rule were significantly associated with higher risks of preterm birth among foreign-born Latinx birthing people. After the leaked draft, the odds of preterm birth among uninsured foreign-born Latinx birthing people increased by 7% compared with privately insured Latinx birthing people (OR: 1.070; 95% CI: 1.026, 1.116). In addition, the odds of preterm births among uninsured foreign-born Latinx birthing people increased by 6.2% compared with privately insured Latinx birthing people (OR: 1.062; 95% CI: 1.016, 1.110). On the other hand, the odds of preterm births among uninsured US-born Latinx birthing people was not significantly associated with the leaked draft or the publication of the proposed public charge rule, showing a 5% level of significance. 

Among the risk factors, mother’s age, education, marital status, birth order, and county median household income were significantly associated with the odds of preterm births. Higher age of birthing person and having the first child were associated with higher rates of preterm births, while higher education of birthing person and higher county-level median household income were associated with lower rates of preterm births among US-born and foreign-born Latinx birthing people.

### 3.3. Event Study Results

Figure 1 shows the quarterly odds of preterm births among uninsured Latinx birthing people relative to privately insured people. The results for US-born Latinx birthing people are reported in the left chart, and the results for foreign-born people are reported in the right chart, while blue dots denote the odds of having preterm births among uninsured Latinx birthing people compared with privately insured, and red lines denote linear fit predicted plots for before and after the changes to the public charge rule. Black vertical lines denote the confidence interval for the odds ratios, and gray dotted vertical lines show the changes to the public charge rule, including the leak of draft executive order in the first quarter of 2017 and the publication of the proposed rule in the last quarter of 2018.

The odds of preterm births among uninsured US-born Latinx birthing people ranged from 1.05 to 1.65 compared with privately insured US-born Latinx birthing people. This shows that uninsured US-born Latinx birthing people were more likely to have preterm births than privately insured people. Before the leaked draft executive order, the predicted linear fit for the odds of preterm births showed a relatively stable trend with fluctuations. After the leaked draft executive order, a jump in the odds of preterm births occurred but showed a gradually decreasing trend with fluctuations after that.

The odds of preterm births among uninsured foreign-born Latinx birthing people ranged from 0.85 to 1.15 compared with privately insured people. This shows that uninsured foreign-born Latinx birthing people were not always more likely to have preterm births than privately insured people. After the leaked draft executive order, a jump in the odds of preterm births also occurred and maintained an elevated level of preterm risk throughout the study period. After the publication of a proposed public charge rule, the odds of preterm births marked the highest spike over the study period.

## 4. Discussion

We found that the leaked draft executive order in 2017 and the publication of the proposed public charge rule in 2018 were significantly associated with higher rates of preterm births among uninsured foreign-born Latinx birthing people in the United States compared with privately insured ones. These findings suggest the changes to the public charge rule during the Trump administration may have been negatively associated with birth outcomes among uninsured foreign-born Latinx birthing people in the United States and therefore put them and their newborns at greater risk. This study confirms the results of previous studies that the risk of having preterm births appears to have significantly increased beyond expected levels after the 2016 presidential election among Latinx birthing people [1,14,15] and further shows that the increased level of preterm births persisted, especially among uninsured foreign-born Latinx birthing people through the changes to the public charge rule during the Trump administration.

Elevated psychosocial stress and anxiety caused by anti-immigrant policies during the 2016 presidential election were suggested as factors associated with the increased preterm births among Latinx birthing people. In addition, inadequate prenatal care utilization after hostile rhetoric against immigrants during the 2016 election was also suggested as a factor associated with elevated preterm births. Existing studies also suggested that immigrant families suffered from anxiety and stress due to immigration-related concerns [16,17] and limited access to care due to Medicaid disenrollment initiated by the changes to the public charge rule [9,10,18,19].

Uninsured foreign-born Latinx birthing people were more likely to be affected by the changes to the public charge rule compared with privately insured foreign-born Latinx birthing people. According to the 2019 Well-Being and Basic Needs Survey, 15.6% of adults in immigrant families avoided noncash public benefit programs due to immigration-related concerns and 45% among them avoid either Medicaid or the CHIP program [19]. Those who disenrolled from Medicaid or CHIP due to immigration-related concerns were more likely to be uninsured, at least in the short term. On the other hand, privately insured Latinx birthing people are less likely to enroll in Medicaid, because employer-based insurance provides better coverage and access to prenatal care than the Medicaid program [20]. In addition, foreign-born Latinx birthing people were more likely to be affected by the public charge rule compared with US-born Latinx birthing people, because the public charge rule can deny permanent residency applications based on previous or potential use of public benefits [8].

We found that uninsured Latinx birthing people were significantly different from privately insured Latinx birthing people, and these findings are consistent with the previous studies that found significant differences between them in terms of age, education, marital status, birth order, county income, county unemployment rate, and risk of preterm birth [21,22,23]. We also found uninsured foreign-born Latinx birthing people showed lower risk of preterm birth than uninsured US-born Latinx birthing people. This finding is also consistent with the existing studies that support the so-called “healthy migrant hypothesis” [24,25,26]. Better childhood nutrition and positive health behavior among selective migrants were suggested as the main reasons for better health outcomes [27,28,29,30]. On the other hand, we found that privately insured US-born Latinx birthing people showed lower risk of preterm births than privately insured foreign-born Latinx birthing people. Advantages in childhood nutrition and positive health behavior among privately insured foreign-born Latinx birthing people might not be enough to overcome other disadvantages in language, cultural barriers, and other socioeconomic factors compared with privately insured US-born Latinx birthing people [31].

The results of this study should be interpreted with caution for the following reasons. First, the study does not cover the period that the final public charge rule was temporarily implemented in from February 2020 to March 2021. The effect of implementing the final public charge rule may be different from the chilling effect of the publication of the suggested public charge rule. However, previous studies found the chilling effect of the changes to the public charge rule on Medicaid disenrollment date since 2018, and the chilling effect on Medicaid disenrollment might also lead to substantial anxiety and stress among uninsured Latinx pregnant people. Second, this study defines Latinx immigrants as the foreign-born Latinx population, including naturalized citizen, permanent residents, and temporary visa holders living in the United States. In theory, naturalized citizens and permanent residents among foreign-born Latinx population were less likely to be affected by the changes to the public charge rule, because the rule is not applicable to citizenship applications and permanent residency renewals [8]. The risk of having preterm births after the changes to the public charge rule among naturalized citizens and permanent residents can be different from that of temporary visa holders. However, the 2019 Well-Being and Basic Needs Survey reported that not only temporary visa holders but also naturalized citizens and permanent residents also avoided public benefits amidst the uncertainty about immigration-related consequences. Third, the changes to the public charge rule are the main exposure of this study, and we assumed that other anti-immigration policies, including executive orders with an anti-immigrant stance signed in January 2017 (E.O. 13767, 13768, 13769), did not have a long-term impact on the risk of preterm births among foreign-born Latinx birthing people. The 2016 election itself and other anti-immigration policies during the Trump administration may have a long-term impact on the risk of preterm births among foreign-born Latinx birthing people. Fourth, this study does not account for variations in the risk of preterm births from country of origin among foreign-born Latinx birthing people. Social and nonsocial factors related to the risk of preterm births may be associated with a particular country among foreign-born Latinx birthing people [32].

## 5. Conclusions

This study indicated a significant increase in preterm births among uninsured foreign-born Latinx birthing people in the United States after the leaked draft executive order in 2017 and the proposed public charge rule announced in 2018. The findings of this study suggest that the changes to the public charge rule may lead to enhanced risk of preterm births among uninsured foreign-born Latinx birthing people. This research contributes to the body of literature by showing the risk of having preterm births was maintained at a higher level among uninsured foreign-born Latinx birthing people during the Trump administration through the changes to the public charge rule. Adverse pregnancy outcomes can have various long-term effects on both the mother’s health and the child’s cognitive development. Lawmakers and policy practitioners should not underestimate the adverse healthcare outcomes of anti-immigration policies.

## Figures and Tables

**Figure 1 healthcare-11-02054-f001:**
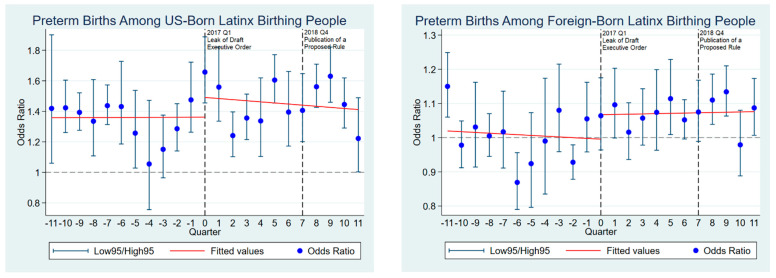
Odds of preterm birth among uninsured Latinx birthing people in the US.

**Table 1 healthcare-11-02054-t001:** Descriptive Statistics of Latinx Birthing People by Nativity, Insurance Status, and Period Related to Public Charge Rule (*n* = 1,692,636) ^a,b^.

	Pre-Leaked Draft Executive Order (*n* = 1,047,740)				Post-Leaked Draft Executive Order (*n* = 517,595)				Post-Publication of a Proposed Rule (*n* = 127,301)			
	US-Born Latinx Birthing People (*n* = 575,956)		Foreign-Born Latinx Birthing People (*n* = 471,784)		US-Born Latinx Birthing People (*n* = 292,220)		Foreign-Born Latinx Birthing People (*n* = 225,375)		US-Born Latinx Birthing People (*n* = 72,119)		Foreign-Born Latinx Birthing People (*n* = 55,182)	
	Uninsured (*n* = 25,311)	Privately Insured (*n* = 550,645)	Uninsured (*n* = 176,837)	Privately Insured (*n* = 294,947)	Uninsured (*n* = 12,401)	Privately Insured (*n* = 279,819)	Uninsured (*n* = 79,617)	Privately Insured (*n* = 145,758)	Uninsured (*n* = 3048)	Privately Insured (*n* = 69,071)	Uninsured (*n* = 19,842)	Privately Insured (*n* = 35,340)
*Outcome measure*												
Preterm Birth	13.40	8.79	10.99	9.18	14.29	9.01	11.46	9.29	15.71	9.61	12.25	10.02
*Mother’s information*												
Age ^d^	25.09	28.72	28.61	31.05	25.65	28.95	28.88	31.15	25.87	29.10	28.97	31.23
Education												
Less than high school	31.78	5.28	54.34	15.80	29.45	4.49	51.81	13.74	25.00	3.95	48.73	12.26
High school grad	34.33	23.56	28.83	26.00	32.88	23.49	28.71	25.93	37.08	23.73	30.87	25.98
Some college	25.36	40.40	8.97	27.50	27.18	39.97	10.22	27.69	27.01	39.18	10.31	27.57
Bachelor’s degree	6.72	21.09	5.88	20.58	8.03	22.02	6.95	21.67	8.36	22.68	7.67	22.89
More than college	1.81	9.66	1.98	10.12	2.46	10.02	2.31	10.98	2.54	10.47	2.42	11.30
Marital status												
Married	40.00	62.66	45.30	75.00	38.93	43.82	44.53	60.43	42.29	44.51	45.08	61.07
Unmarried	60.00	37.34	54.70	25.00	61.07	56.18	55.47	39.57	57.71	55.49	54.92	38.93
*Infant’s information*												
Firstborn	41.97	42.92	28.85	35.52	39.44	42.88	26.94	36.65	39.76	43.80	27.32	37.76
Second-born or more	58.03	57.08	71.15	64.48	60.56	57.12	73.06	63.35	60.24	56.20	72.68	62.24
*County Characteristics*												
Income ^d^	53,876	59,224	53,041	60,760	57,196	63,062	56,674	64,369	60,638	66,620	60,091	67,781
Unemployment ^d^	8.12	8.43	8.10 ^c^	8.11 ^c^	6.45	6.70	6.38 ^c^	6.37 ^c^	5.68	5.87	5.58	5.54

^a^ The CT, NJ, and RI residents were excluded in the sample, because the states started implementing the 2003 revision after January 2014. ^b^ The main exposures of this study are the changes to the public charge rule, which include the leaked draft executive order in January 2017 and the publication of a suggested public charge policy in October 2018. The study period consists of three periods: (1) January 2014 to December 2016; (2) January 2017 to October 2018; and (3) November 2018 to December 2019. ^c^ Independent sample T test shows the statistically significant differences between uninsured and privately insured Latinx birthing people for all variables except county unemployment rate for foreign-born Latinx birthing people during pre- and post-leaked draft executive order periods (January 2014–October 2018). ^d^ Means are reported for all continuous variables. Percentages are reported for all indicator variables.

**Table 2 healthcare-11-02054-t002:** The Association of the Publication of a Proposed Public Charge Rule in 2018 on the Risk of Preterm Births among US-Born and Foreign-Born Latinx Populations in the United States ^a,b,d^.

	US-Born Latinx Birthing People		Foreign-Born Latinx Birthing People	
	O.R.	C.I.	O.R.	C.I.
Uninsured	1.353 ***	[1.207, 1.516]	1.013	[0.938, 1.094]
Leaked Draft ^c^	1.094 ***	[1.051, 1.139]	1.052 *	[0.999, 1.109]
Proposed Rule ^c^	1.093 ***	[1.058, 1.129]	1.062 ***	[1.019, 1.106]
Uninsured X Leaked Draft ^c^	1.079	[0.984, 1.185]	1.070 ***	[1.026, 1.116]
Uninsured X Proposed Rule ^c^	1.117 *	[0.983, 1.269]	1.062 ***	[1.016, 1.110]
County Income	0.766 ***	[0.701, 0.837]	0.780 **	[0.630, 0.966]
County Unemployment	1.005	[0.992, 1.019]	1.000	[0.982, 1.019]
Mother’s age	1.033 ***	[1.031, 1.036]	1.025 ***	[1.020, 1.030]
Highschool	0.774 ***	[0.747, 0.802]	0.876 ***	[0.842, 0.913]
Some College	0.712 ***	[0.695, 0.730]	0.816 ***	[0.788, 0.844]
Bachelor	0.580 ***	[0.555, 0.605]	0.689 ***	[0.651, 0.729]
Master or more	0.550 ***	[0.525, 0.576]	0.636 ***	[0.589, 0.687]
Married	0.833 ***	[0.807, 0.859]	0.850 ***	[0.836, 0.864]
First-born	1.127 ***	[1.060, 1.198]	1.067 **	[1.003, 1.134]

^a^ The treatment group includes uninsured Latinx birthing people and the comparison group is privately insured Latinx birthing people living in the United States. The results for US-born Latinx birthing people are reported in the left column, and the results for foreign-born Latinx birthing people are reported in the right column. ^b^ Odds ratios are reported for all variables. Confidence intervals are in parentheses. ^c^ The main exposures of this study are the changes to the public charge rule, which include the leaked draft executive order regarding the changes to public charge policy in January 2017 and the publication of a suggested public charge policy in October 2018. ^d^ Year and state fixed effects were included but not reported. * *p* < 0.1 percent; ** *p* < 0.05 percent; *** *p* < 0.01 percent.

## Data Availability

The dataset generated during the current study is not publicly available as it contains protected Health Information. Information on how to obtain it and reproduce the analysis is available from the corresponding author on request.

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
