# Peer review of "The Association of the Publication of a Proposed Public Charge Rule with Preterm Births among Uninsured Foreign-Born Latinx Birthing People in the United States"

_healthcare, 2023, doi:10.3390/healthcare11142054_

Round 1

Reviewer 1 Report

Overall, the manuscript provides the results of a study investigating the effects of changes to the Public Charge Rule on the likelihood of premature birth among the Latinx population in the United States. Overall, the article is written well in a well-elaborated manner with relevant references. However, there are a few areas where the manuscript can be improved:

  1. Introduction: The introduction could be strengthened by providing more background information on the Public Charge Rule and its potential impact on healthcare access and outcomes for immigrant populations. This would help readers understand the significance of the research topic. For example, one of the articles mentioned this concisely and clearly, "The proposed rule instructs immigration officials to consider a broadened array of public benefits—including (nonemergency) Medicaid, Supplemental Nutrition Assistance Program (SNAP), Medicare Part D low-income subsidies, and housing assistance such as Section 8 housing vouchers—along with other factors when making public charge determinations. In response to the proposed changes, many immigrant parents are expected to disenroll themselves and their children from safety-net benefits programs." The author should consider writing something similar and cite accordingly. 

  2. Ethical Considerations: Whether IRB from the author's university is enough, should we get the IRB exempt from the NCHS Research Ethics Review Board (ERB) Approval? Also, the author should mention that IRB was exempted because it analyzed publicly available deidentified data.

  3. Analysis: The manuscript briefly mentions the use of multivariate logistic regression models, but it does not provide details about the specific variables included in the models or the statistical methods used to assess the associations.  

  4. Results: The results section presents descriptive statistics. However, it would be helpful to include effect size estimates, confidence intervals, and p-values to determine the statistical significance of the findings. This would enhance the interpretation and credibility of the results.

  5. The manuscript would benefit from a clear conclusion section that summarizes the main findings and their implications. This would provide closure to the study and reinforce the significance of the research.

  6. Writing Style and Clarity: The manuscript could benefit from improved clarity and organization. The writing style is dense and contains lengthy sentences, making it challenging to follow the flow of the text. Breaking down complex ideas into shorter sentences and paragraphs would enhance readability. The manuscript would benefit from proofreading and editing for clarity, grammar, and formatting.

Addressing these points would strengthen the manuscript and make it more informative and understandable to readers.

The manuscript could benefit from improved clarity and organization. The writing style is dense and contains lengthy sentences, making it challenging to follow the flow of the text. Breaking down complex ideas into shorter sentences and paragraphs would enhance readability. The manuscript would benefit from proofreading and editing for clarity, grammar, and formatting.

Author Response

Thank you for giving me the opportunity to submit a revised draft of my manuscript titled "Preterm Births Significantly Increased after the Publication of a Proposed Public Charge Rule among Foreign-born Uninsured Latinx Birthing People in the United States.” to Healthcare. I am grateful to the reviewers for their insightful comments on my paper. I have been able to incorporate changes to reflect all suggestions provided by the reviewers. I have highlighted the changes within the manuscript. For some reasons, the line numbers for pages 1-3 were not available. I couldn’t add line numbers for the responses related to those pages.

Reviewer 1

Comments and Suggestions for Authors

Overall, the manuscript provides the results of a study investigating the effects of changes to the Public Charge Rule on the likelihood of premature birth among the Latinx population in the United States. Overall, the article is written well in a well-elaborated manner with relevant references. However, there are a few areas where the manuscript can be improved:

  1. Introduction: The introduction could be strengthened by providing more background information on the Public Charge Rule and its potential impact on healthcare access and outcomes for immigrant populations. This would help readers understand the significance of the research topic. For example, one of the articles mentioned this concisely and clearly, "The proposed rule instructs immigration officials to consider a broadened array of public benefits—including (nonemergency) Medicaid, Supplemental Nutrition Assistance Program (SNAP), Medicare Part D low-income subsidies, and housing assistance such as Section 8 housing vouchers—along with other factors when making public charge determinations. In response to the proposed changes, many immigrant parents are expected to disenroll themselves and their children from safety-net benefits programs." The author should consider writing something similar and cite accordingly. 

Following the comment, I provided more background information about the changes to the Public Charge Rule to read: “The Trump administration broadened the scope of the rule to incorporate cash and noncash benefits including the Supplemental Nutrition Assistance Program (SNAP), nonemergency Medicaid, Section 8 housing benefits/public housing, and drug subsidies under Medicare Part D [8,9,10]. The changed rule will make public charge determination by considering not only previous use of public benefits but also the applicant’s potential use of public benefits, which will be evaluated based on the applicant’s personal attribute such as age, income, education, and English proficiency [9, 10]. The changes to the rule were anticipated to make immigrant families to avoid public benefits due to immigration related concerns [9,10].”

  1. Ethical Considerations: Whether IRB from the author's university is enough, should we get the IRB exempt from the NCHS Research Ethics Review Board (ERB) Approval? Also, the author should mention that IRB was exempted because it analyzed publicly available deidentified data.

Following the comment, I revised the ethical consideration to read: “The reviewers from the National Center for Health Statistics approved this study as appropriate request and the Institutional Review Board (IRB) of the author’s university exempted this study from IRB oversight because this study analyzed publicly available deidentified data (STUDY ID: 00021733).”

  1. Analysis: The manuscript briefly mentions the use of multivariate logistic regression models, but it does not provide details about the specific variables included in the models or the statistical methods used to assess the associations.  

Thank you for the reviewer’s comment. The analysis employed a difference-in-differences approach that compared the differences in preterm birth before and after the changes to the Public Charge Rule among uninsured Latinx birthing parents in relation to privately-insured parents by using multivariate logistic regression models. Following the comment, I provide the specific variables included in the multivariate logistic regression models to read: “Outcomeist is an indicator variable for preterm birth;  is an identifier for uninsured Latinx birthing people,  and are parameters that measure the impact of the changes to Public Charge Rule on uninsured Latinx birthing people compared to privately insured Latinx birthing parent. Xist is a vector of birthing person–infant dyad characteristics including mother’s age, education, marital status, and infant’s birth order and county characteristics including median household income and unemployment rate. μs and θt are vectors of state and year fixed effects, and εist is an error term.”

  1. Results: The results section presents descriptive statistics. However, it would be helpful to include effect size estimates, confidence intervals, and p-values to determine the statistical significance of the findings. This would enhance the interpretation and credibility of the results.

I agree with the reviewer’s comment. Unfortunately, I couldn’t add any other values into the descriptive statistics due to limited page space. Table 1 compared the characteristics of Latinx birthing people by nativity, insurance status and period related to Public Charge Rule. Adding one more column for each group was not possible, while keeping it readable on a page. I conducted independent sample T test for all variables between uninsured and privately-insured Latinx birthing people and the test results show the statistically significant differences between uninsured and privately-insured Latinx birthing people for all variables except county unemployment rate for foreign-born Latinx birthing people during pre and post leaked draft executive order periods (Jan. 2014 - Oct. 2018). I mentioned this in a note of Table 1.

  1. The manuscript would benefit from a clear conclusion section that summarizes the main findings and their implications. This would provide closure to the study and reinforce the significance of the research.

Following the comment, I added the conclusion section to read: “This study indicated a significant increase in preterm births among uninsured foreign-born Latinx birthing people in the United States after the leaked draft executive order in 2017 and the proposed Public Charge Rule announced in 2018. The findings of this study suggest that the changes to the Public Charge Rule may lead to enhanced risk of preterm birth among uninsured foreign-born Latinx birthing people. This research contributes to the body of literature by showing the risk of having preterm birth had been maintained at a higher level among uninsured foreign-born Latinx birthing people during the Trump administration through the changes to the Public Charge Rule. Ad-verse pregnancy outcomes can have various long-term effects on both the mother’s health and the child’s cognitive development. Lawmakers and policy practitioners should not underestimate the adverse healthcare outcomes of anti-immigration policies.” (Lines 216 - 227)

  1. Writing Style and Clarity: The manuscript could benefit from improved clarity and organization. The writing style is dense and contains lengthy sentences, making it challenging to follow the flow of the text. Breaking down complex ideas into shorter sentences and paragraphs would enhance readability. The manuscript would benefit from proofreading and editing for clarity, grammar, and formatting.

Following the comment, the manuscript was revised by proofreading and editing for clarity, grammar, and formatting.

Addressing these points would strengthen the manuscript and make it more informative and understandable to readers.

Comments on the Quality of English Language

The manuscript could benefit from improved clarity and organization. The writing style is dense and contains lengthy sentences, making it challenging to follow the flow of the text. Breaking down complex ideas into shorter sentences and paragraphs would enhance readability. The manuscript would benefit from proofreading and editing for clarity, grammar, and formatting.

Following the comment, the manuscript was revised by proofreading and editing for clarity, grammar, and formatting.

Reviewer 2 Report

I would like to thank the author for the effort to look into the matter of birth outcomes by maternal country of birth among Latinx Birthing People in the United States. The data on the topic are scant, and research on the subject matter is welcome. A few specific suggestions and observations are below. Line numbers do not appear in the available version of the draft until section 3.2. Difference-in-Differences Results, hence the first reference is to a page number/paragraph.

Article Title: the title captures the essence of the research question. A suggestion is to reformulate it to sound a title of an academic article, rather than a statement.

Introduction: p. 3, para 3: “The changes to the Public Charge Rule may engender elevated levels of psychosocial stress and anxiety, which could adversely impact pregnant individuals in the Latinx community. Nonetheless, there is a paucity of research investigating the effect of the changes to the Public Charge Rule on the likelihood of premature birth among the Latinx population. The objective of this study is to evaluate effects of the changes to the Public Charge Rule on the susceptibility to premature birth among the Latinx population in the United States.“ – I appreciated the acknowledgement of the paucity research on the subject matter, and the link thereof in the formulation of the research objective.  

4. DISCUSSION seems to confirm the findings of other studies, according to which the 2016 US presidential election [appears to have been] associated with an increase in preterm births among US Latinx.

The section following the acknowledgement that “the results of this study should be interpreted with caution” with the list of reasons is appreciated.

One suggestion is to break down the discussion section into discussion and conclusion.

Author Response

Thank you for giving me the opportunity to submit a revised draft of my manuscript titled "Preterm Births Significantly Increased after the Publication of a Proposed Public Charge Rule among Foreign-born Uninsured Latinx Birthing People in the United States.” to Healthcare. I am grateful to the reviewers for their insightful comments on my paper. I have been able to incorporate changes to reflect all suggestions provided by the reviewers. I have highlighted the changes within the manuscript. For some reasons, the line numbers for pages 1-3 were not available. I couldn’t add line numbers for the responses related to those pages.

Reviewer 2

Comments and Suggestions for Authors

I would like to thank the author for the effort to look into the matter of birth outcomes by maternal country of birth among Latinx Birthing People in the United States. The data on the topic are scant, and research on the subject matter is welcome. A few specific suggestions and observations are below. Line numbers do not appear in the available version of the draft until section 3.2. Difference-in-Differences Results, hence the first reference is to a page number/paragraph.

  1. Article Title: the title captures the essence of the research question. A suggestion is to reformulate it to sound a title of an academic article, rather than a statement.

Following the comment, the title was reformulated to “The Association of the Publication of a Proposed Public Charge Rule with Preterm Births among Foreign-born Uninsured Latinx Birthing People in the United States.”

  1. Introduction: p. 3, para 3: “The changes to the Public Charge Rule may engender elevated levels of psychosocial stress and anxiety, which could adversely impact pregnant individuals in the Latinx community. Nonetheless, there is a paucity of research investigating the effect of the changes to the Public Charge Rule on the likelihood of premature birth among the Latinx population. The objective of this study is to evaluate effects of the changes to the Public Charge Rule on the susceptibility to premature birth among the Latinx population in the United States.“ – I appreciated the acknowledgement of the paucity research on the subject matter, and the link thereof in the formulation of the research objective.  

I also appreciated the reviewer’s comment.

  1. DISCUSSION seems to confirm the findings of other studies, according to which the 2016 US presidential election [appears to have been] associated with an increase in preterm births among US Latinx.

Following the comment, the wording was revised to “This study confirms the results of previous studies that the risk of having preterm birth appears to have significantly increased beyond expected levels after the 2016 presidential election among Latinx birthing people [1,12,13].” (Lines 107 - 109)

  1. The section following the acknowledgement that “the results of this study should be interpreted with caution” with the list of reasons is appreciated.

Following the comment, I added the reason why country of origin among foreign-born Latinx birthing people may influence the risk of having preterm birth. Social and non-social factors related to the risk of preterm birth may be associated with a particular country among foreign-born Latinx birthing people [30]. (Lines 213 – 214)

  1. One suggestion is to break down the discussion section into discussion and conclusion.

Following the comment, I added the conclusion section to read: “This study indicated a significant increase in preterm births among uninsured foreign-born Latinx birthing people in the United States after the leaked draft executive order in 2017 and the proposed Public Charge Rule announced in 2018. The findings of this study suggest that the changes to the Public Charge Rule may lead to enhanced risk of preterm birth among uninsured foreign-born Latinx birthing people. This research contributes to the body of literature by showing the risk of having preterm birth had been maintained at a higher level among uninsured foreign-born Latinx birthing people during the Trump administration through the changes to the Public Charge Rule. Ad-verse pregnancy outcomes can have various long-term effects on both the mother’s health and the child’s cognitive development. Lawmakers and policy practitioners should not underestimate the adverse healthcare outcomes of anti-immigration policies.” (Lines 216 - 227)